# You Have Got a Fast CAR: Chimeric Antigen Receptor NK Cells in Cancer Therapy

**DOI:** 10.3390/cancers12030706

**Published:** 2020-03-17

**Authors:** Aline Pfefferle, Nicholas D. Huntington

**Affiliations:** 1Biomedicine Discovery Institute and the Department of Biochemistry and Molecular Biology, Monash University, Clayton, VIC 3800, Australia; nicholas.huntington@monash.edu; 2oNKo-Innate Pty Ltd., Clayton, VIC 3800, Australia

**Keywords:** natural killer cells, chimeric antigen receptors, immunotherapy

## Abstract

The clinical success stories of chimeric antigen receptor (CAR)-T cell therapy against B-cell malignancies have contributed to immunotherapy being at the forefront of cancer therapy today. Their success has fueled interest in improving CAR constructs, identifying additional antigens to target, and clinically evaluating them across a wide range of malignancies. However, along with the exciting potential of CAR-T therapy comes the real possibility of serious side effects. While the FDA has approved commercialized CAR-T cell therapy, challenges associated with manufacturing, costs, and related toxicities have resulted in increased attention being paid to implementing CAR technology in innate cytotoxic natural killer (NK) cells. Here, we review the current landscape of the CAR-NK field, from successful clinical implementation to outstanding challenges which remain to be addressed to deliver the full potential of this therapy to more patients.

## 1. Background of Chimeric Antigen Receptors

In 1986, Steven Rosenberg harnessed the potential of tumor-specific T cells to treat melanoma patients, starting a new chapter of cell-based therapy for cancer patients [1]. While the process of harnessing tumor-infiltrating lymphocytes (TIL) is logistically challenging, generating and expanding T cells specific to a patient’s neo-antigen is equally challenging. Therefore, TIL-based therapy has major limitations and is a perfect example of patient-specific treatment [2]. An alternative to TIL-based therapy is to genetically engineer the T cell receptor (TCR) to confer specificity to a particular tumor target, aptly named a chimeric antigen receptor (CAR). Although CAR therapy is still patient-specific, due to the use of T cells which need to be human leukocyte antigen (HLA)-matched, the CAR construct itself is tumor-specific [3]. The ground-breaking work of redirecting T cell specificity through genetic engineering was performed at the Weizmann Institute by Zelig Eshhar, who went on to develop a first-generation CAR at the National Institute of Health in Steven Rosenberg’s lab [4,5]. The addition of co-stimulatory domains to improve the potency of CAR signaling was developed by Michel Sadelain, giving rise to second- and third-generation CAR constructs (Figure 1a) [6].

CAR constructs are made up of four main components: the single-chain variable fragment (scFv), the hinge, the transmembrane domain (TM), and the intracellular signaling domains. Each of the four components has been studied and optimized in T cells to maximize tumor detection, T cell activation, and tumor elimination [7].

### 1.1. Ectodomains

The scFv, the antigen binding component conferring specificity, consists of a heavy (V_H_) and light (V_L_) variable fragment connected by a flexible linker. The order of the variable fragments, as well as the length of the linker, have been shown to affect the antigen binding affinity and stability of the construct. Both the epitope location and abundancy need to be considered when designing the scFv [8]. 

The length of the hinge region is important for the formation of the immune synapse. Depending on the antigen distance from the cell surface, the hinge length needs to be adjusted to allow for an optimal distance between the effector and target cell. A longer hinge provides increased flexibility to reach membrane-proximal antigens, while shorter hinges suffice for membrane-distal antigens [9,10]. Amino acid sequences from CD28 or CD8α are commonly used, as well as CH2 and CH3 domains from IgG1, 2, or 4. While both CH2 and CH3 domains can also be utilized to detect CAR expression on the cell’s surface, binding between CH2 domains and Fcγ receptors has been observed and can lead to off-target activation [8,11]. 

### 1.2. Transmembrane Region

The transmembrane (TM) domain consists of a hydrophobic alpha helix that spans the cell membrane and anchors the CAR construct. While CD8α and CD28 are most commonly used to date, the choice of TM domain has been shown to affect the functionality of the CAR construct mediated through the degree of cell activation. A CD28-derived TM domain was more prone to induce activation-induced cell death (AICD) in T cells compared to a CD8α-derived TM domain, while a CD3ζ-derived TM domain facilitated CAR dimerization with endogenous TCRs, leading to T cell activation [12,13]. 

### 1.3. Endodomains

The evolution of the CAR construct has primarily focused on optimizing the intracellular signaling domains, with the first three generations of CAR constructs referring to the number of activating and co-stimulatory molecules making up the endodomain. The first-generation CAR constructs only contained an activating domain, namely CD3ζ, whilst the second- and third-generation constructs contain one and two co-stimulatory domains, respectively. In T cells, domains from both the CD28 family (CD28 and ICOS) and the TNF receptor family (4-1BB, OX40, and CD27) have been widely tested, with the FDA-approved CD28- and 4-1BB-containing constructs being most commonly used to date [14,15,16,17,18]. The choice of co-stimulatory domains allows for fine-tuning of the desired T cell response, whereby CD28-based CARs exhibit an increased cytolytic capacity and shorter persistence compared to 4-1BB-based CARs [19]. 

### 1.4. Limitations of CAR-T Cells

Successful clinical trials with second-generation autologous CD19-targeted CAR-T cells resulted in FDA approval, with two products, namely Kymriah^®^ (Novartis) and Yescarta^®^ (Kite/Gilead), currently on the market [14,15]. The manufacturing time and costs, as well as severe toxicities, associated with CAR-T cell therapies are current limitations for these products [20]. Generating an autologous CAR-T cell product from chemo-refractory patients is severely limited by the quality of T cells obtained, as well as the survival time of the patients themselves, as the manufacturing process takes 2–4 weeks (Figure 2). Additionally, the patients need to be treatment-free for two weeks prior to apheresis to ensure sufficient cell numbers and viability for the manufacturing process [21]. However, the alternative—treatment with an allogeneic CAR-T cell product—always carries the serious risk of inducing graft-versus-host disease (GvHD) [22]. Irrespective of whether the treatment is autologous or allogeneic, serious side effects are commonly observed with CAR-T cell therapy. Neurotoxicity, immune effector cell-associated neurologic syndrome (ICANS), and cytokine release syndrome (CRS) are frequently observed, both of which are serious conditions which can be fatal if not treated, particularly in the case of CRS (Figure 3) [23,24]. It has been shown that 37–93% of patients receiving CAR-T cell therapy for leukemia or lymphoma develop CRS with symptoms ranging from a high fever to hypotension, hypoxia, and tachycardia, with high-grade CRS requiring intensive care treatment [20,21,25,26,27,28]. CRS is the result of on-target CAR-T cell activation leading to massive proliferation and cytokine/chemokine release, with elevated levels of IL-6, IFNγ, GM-CSF, and soluble IL-6R and IL-2Rα being commonly observed. Tocilizumab, which is an anti-IL-6 receptor agonist, is routinely used to counter the high levels of IL-6, with corticosteroids being administered in patients not responding to tocilizumab [14,21,26,29,30,31,32,33]. ICANS often develop after CRS, with symptoms ranging from encephalopathy to expressive aphasia and seizures, but rare cases of cerebral edema have also been reported [26,34,35]. Between 23% and 67% of patients treated with CAR-T therapy against lymphoma or leukemia develop ICANS [20,21,25,26,27,28]. CAR-T cell-induced toxicities usually require three weeks of ‘in-patient’ treatment. 

The fourth generation of CAR constructs is focused on addressing the current limitations of CAR-T cell therapy. These next-generation CAR designs can be further classified into subgroups based on their rational (Figure 1b). ON-switch CARs rely on a small molecule to assemble the fragmented CAR construct, allowing for controlled CAR activation through the administration of a drug [36]. Universal CARs are another type of fragmented CAR design, whereby the antigen-specific portion can be exchanged to facilitate the targeting of numerous different cancer types through the same TM and intracellular signaling construct [37,38,39,40]. OR-gate CARs aim to prevent tumor escape by providing two scFv domains against different targets, which are either bound to a single TM and intracellular domain (Tandem CAR) or are simply two complete CAR constructs expressed on the same cell (Dual CAR) [41,42,43]. Signaling through either scFv will activate the T cell. AND-gate CARs, as the name implies, also feature two scFvs, but require the presence of both antigens on the same cell before signal propagation. This approach allows for the targeting of non-tumor specific antigens, as tumor specificity is achieved by the dual expression of both antigens (Combinatorial CAR and synNotch Receptor) [44,45,46]. T cells redirected for universal cytokine killing (TRUCK)—CAR constructs carrying a transgenic ‘payload’are a novel design for targeting solid tumors. These constructs feature the CAR-inducible expression of a transgene product, such as IL-12, facilitating release of the ‘payload’ at the tumor site to modulate the tumor microenvironment [47,48]. Similarly, inhibitory CARs are focused on turning an immunosuppressive signal from a tumor cell into an activating signal by fusing the extracellular inhibitory domain, for example, PD-1, to an activating intracellular CAR domain [49]. 

Lastly, in an attempt to increase the safety profile of CAR-T cells, a suicide switch that can be triggered upon the development of adverse effects was developed, yet only ~10% of current clinical trials implement this safety feature [50]. One alternative strategy for mitigating the limitations of CAR-T cell-based therapy is to utilize natural killer (NK) cells instead. 

### 1.5. Advantages of CAR-NK Cells 

Adoptive cell therapy (ACT) with allogeneic haplo-identical NK cells has been clinically proven to be safe, without the risk of inducing GvHD. NK cells have been shown to be major contributors to the graft-versus-tumor (GvT) response observed after hematopoietic stem cell transplantation (HSCT) for acute myeloid leukemia (AML) [51,52]. Tumor escape via the loss of antigen or via the loss of major histocompatibility complex I (MHC-I) expression renders CAR-T cells helpless in detecting tumor cells [53]. CAR-NK cells, on the other hand, retain their innate cytolytic capacity against germline-encoded tumor/stress ligands and are able to detect MHC-I-negative tumor cells due to their lack of a self-antigen [11,54]. Decreased MHC-I expression is a characteristic ascribed to cancer stem cells (CSCs), which, together with the ligand expression of activating receptors, namely NKp30, NKp44, and NKG2D, is believed to sensitize CSCs to cytokine-activated NK cell-mediated killing [55,56,57]. Shedding of the NKG2D ligands MICA and MICB by breast CSCs and the absence of NKG2D ligands on leukemia stem cells are effective mechanisms utilized by CSCs to evade NK cell detection [58,59]. Furthermore, NK cells have the ability to perform serial killing and although generally short-lived, cytomegalovarius (CMV)-induced memory-like adaptive NK cells have been shown to be long-lived and highly potent [60]. The safety profile of NK cells, combined with their anti-tumor potential, makes them a promising cell type for the implementation of CAR technology, which can redirect their cytotoxic potential towards a specific target. The feasibility of manufacturing an off-the-shelf CAR-NK cell product to universally treat patients would significantly increase the speed of administration, effectively reducing the lag time from the decision to treat and first dosing to 1 day (Figure 2). As no severe toxicities are observed or expected with CAR-NK cells, treatment can be administered with ‘out-patient’ follow-up monitoring, significantly reducing the huge indirect costs associated with CAR-T cell therapy due to lengthy post-treatment hospitalization [61]. While an induced pluripotent stem cell (iPSC)-derived CAR-T cell product can also be manufactured as an off-the-shelf product, extra genetic modification is required to remove the endogenous TCR in order to produce a universal product without the need for HLA-matching. Nonetheless, the serious side effects observed from CAR-T cell therapy would not be eliminated through the use of iPSC-derived CAR-T cells, retaining the need for post-treatment hospitalization. 

A recent meta-analysis of ongoing clinical trials across the globe identified 520 active trials examining a total of 64 different CARs, with 96.4% of trials using CAR-T cells [50]. Hence, the CAR-NK cell field is still in its infancy in terms of translating laboratory studies into a clinical setting at present.

## 2. Evolution of the CAR-NK Field 

Unlike T cells, NK cells do not express antigen-specific surface receptors to discriminate malignant cells. Instead, a large repertoire of germline-encoded inhibitory and activating receptors provides the cell with the necessary information to distinguish a healthy cell from a foreign, infected, or transformed cell. It is the balance of signals received, as well as the absence of important inhibitory signals, termed ‘missing self’, that modulate the release of cytotoxic granules for target elimination [54]. Within the blood, cytotoxic CD56^dim^ NK cells and immunomodulatory CD56^bright^ NK cells make up the two main populations of NK cells. In reality, it has been estimated that up to 10^5^ subsets of NK cells exist based on the vast repertoire of surface receptor expression [62,63]. Compared to T cells, NK cells are fairly resistant to genetic engineering and more difficult to expand to the large numbers needed for infusion, as the number of cell divisions they can undergo is limited. To mitigate these challenges, many early CAR-NK cell studies have been performed with NK cell lines instead of primary NK cells obtained from umbilical cords or peripheral blood (Figure 4) [64,65]. 

### 2.1. NK Cell Line (NK-92)

The majority of CAR-NK cell studies to date have been performed with NK-92 cells, including a large number of current clinical trials (Table 1) [50]. Although NK-92 cells phenotypically resemble the immunomodulatory and poorly cytotoxic CD56^bright^ population found in the blood, they are functionally highly cytotoxic [66,67,68]. Their activating receptor repertoire includes NKp30, NKp46, and NKG2D, and their inhibitory repertoire is limited to ILT-2, NKG2A, and KIR2DL4 at low levels [66,68]. The lack of inhibitory killer immunoglobulin-like receptor (KIR) expression on the cell surface and inability to kill through antibody-dependent cellular cytotoxicity (ADCC) due to low or absent CD16 expression greatly differentiates this cell line from primary NK cells. These deficiencies have led to NK-92 cells being genetically engineered to express a high-affinity CD16 receptor, with the rational of combining CD16-expressing NK-92 cells together with anti-tumor monoclonal antibodies to facilitate tumor elimination via ADCC [69]. Furthermore, numerous studies have overserved increased IFNγ release in CAR-expressing NK-92 cells compared to relatively low production in unmodified NK-92 cells in response to target exposure [70,71,72,73,74,75]. 

Developing a cell-line-based CAR product brings the advantage of an unlimited proliferative capacity and reduced sensitivity to repeated freeze/thaw cycles. These properties allow for the manufacturing of a universal off-the-shelf product, severely reducing both the manufacturing time and cost. Conversely, a cell-line-based product raises new questions of safety, as NK-92 cells are both Epstein-Barr virus (EBV)-positive and derived from an NK lymphoma. Therefore, NK-92 cells carry many genetic abnormalities and have the potential to permanently engraft upon infusion, making irradiation prior to infusion an absolute necessity [76]. Numerous studies have demonstrated that NK-92 cells retain their cytotoxicity post-irradiation, but the lack of in vivo proliferation results in clearance of the infused cells after only 7 days [74,77,78,79,80]. Treatment with an NK-92-based CAR product will therefore most likely require multiple rounds of infusion [78]. 

Recently, NK-92 cells were engineered to express a fourth-generation CAR construct originally designed for T cells, termed UniCAR [81]. This CAR platform combines both the universal aspect and the ON-switch design. The two-component platform consists of a CAR construct against the E5B9 epitope, which is not expressed ectopically, and a bi-specific target molecule, namely the E5B9 epitope fused to a tumor-specific scFv. The target molecule provides tumor specificity and can be exchanged to facilitate the targeting of different tumor types. Importantly, the CAR construct is only active in the presence of the target molecule which is individually administered, and due to its limited half-life, functions as an off-on switch. By implementing this CAR design in NK-92 cells, the resulting product is not only universal and inducible, but furthermore, can be manufactured as an off-the-shelf product [81]. 

### 2.2. Cord Blood NK Cells 

NK cells isolated from cord blood (CB) present another possible starting material for a CAR-NK product [82]. Between 15% and 30% of CB lymphocytes are NK cells, and although contradictory reports have been published, they are generally considered to be more naïve in phenotype and function compared to peripheral blood (PB) NK cells. Reduced expressions of CD16, granzyme B, perforin, and KIRs have been reported, as well as a decreased expression of cell adhesion molecules, including CD2, CD11a, CD18, and CD62L. Limited ADCC potential and limited functional maturation through KIR expression, together with a reduction in adhesion molecules, all contribute to the reduced cytotoxic capacity observed in this population of cells [83,84,85,86]. A higher percentage of CB NK cells are CD56^bright^, contributing to the overall higher expression of the inhibitory receptor NKG2A, the ligand for which, HLA-E, is commonly upregulated on tumor cells and is associated with a poor outcome [87]. Therefore, NKG2A is an important inhibitory checkpoint which has been targeted with monoclonal antibody therapy to unleash NK cell potential in the setting of head and neck squamous cell carcinoma [88]. In line with their immature nature, CB NK cells have a higher proliferative capacity and are very receptive to cytokine stimulation. While umbilical cord blood banks make CB-derived NK cells a feasible starting product, the small volume obtained from cord blood requires the NK cells to undergo a large number of cell divisions before obtaining sufficient cell numbers required for infusion [83]. Nonetheless, clinical CAR-NK cell products have been manufactured from CB NK cells and are currently being tested in the clinic [89]. 

### 2.3. Peripheral Blood NK Cells

Approximately 90% of peripheral blood NK cells are cytotoxic CD56^dim^ NK cells, expressing a vast repertoire of inhibitory and activating receptors. Compared to CB NK cells, PB NK cells are more mature, resulting in an increased functionality, but reduced proliferative capacity [90]. Their safety as an adoptive cell therapy product has been proven through numerous clinical trials, in both an autologous and allogeneic setting, with NK cells from matched and HLA-mismatched, as well as matched and KIR-ligand mismatched, donors [51,91,92,93]. The possibility of using cells from an unrelated HLA-mismatched donor substantially increases the number of possible donors to choose from, and therefore, can significantly increase the quality of the final product. Apheresis, a method by which nucleated cells are extracted from the blood of healthy donors, facilitates the isolation of a sufficient number of NK cells, often via CD3 and CD19 depletion, which are then often further expanded and primed by cytokines, such as IL-2 or IL-15, prior to infusion. Cytokine-based and feeder-based expansion protocols will be discussed in Section 3.5 and Section 3.6, respectively. While clinical results have been promising against hematopoietic malignancies, largely AML, the clinical response has been patient-specific and difficult to predict based on pre-clinical in vitro studies. 

### 2.4. iPSC-Derived NK Cells

A fourth, and fairly recent, source of NK cells is iPSC-derived NK (iNK) cells [94]. iPSC-derived NK cells are an ideal source for an off-the-shelf CAR-NK cell product due to their unlimited proliferative potential [95]. Comparatively, the transduction efficiency plays an important role in manufacturing CB or PB-derived CAR-NK cells, and only a single CAR-transduced iPSC cell is needed to generate a universal off-the-shelf CAR-NK cell product. However, iPSC-derived NK cells are still in their infancy and a number of challenges still need to be overcome to generate the ideal off-the-shelf product. iPSCs are commonly derived from non-hematopoietic cells, such as fibroblasts, yielding iNK cells with an immature phenotype characterized by low CD16 expression, high NKG2A expression, and lower KIR expression compared to PB NK cells [94,96,97]. This raises the question of how their cytolytic potential is maintained and released upon target cell encounter, which could be addressed through the expression of a CAR construct. Similar to NK-92 cells, genetically engineering iNK cells to express lacking surface proteins, such as CD16, is a potential solution to this current limitation. Although their proliferative capacity is unmatched, the question of persistence upon infusion remains and needs to be addressed before the true potential of iPSC-derived NK cells can be harnessed [75]. The first iPSC-derived CAR-NK cell product, FT596, expresses a CD19-CAR construct and a high-affinity, non-cleavable CD16 Fc receptor plus an IL-15 receptor fusion protein and will be used in combination with CD20-directed monoclonal antibody therapy. This product therefore addresses not only the limited persistence, but also the low CD16 expression of iNK cells and puts combination immunotherapy to the test. Promising preclinical results from in vivo studies in humanized mice with CD19+ lymphoma have paved the way for a recently listed Phase I clinical trial (NCT04245722). 

## 3. Unique Challenges of CAR-NK Cells

The successful implementation of CAR-NK cells as a therapeutic cancer treatment is currently hampered by three main factors, namely, (1) the resistance of NK cells to genetic engineering, (2) the limited proliferative potential of NK cells, and (3) their limited persistence upon infusion. 

### 3.1. Viral Transduction 

Historically, the viral transduction of primary NK cells with either retrovirus or lentivirus has had poor efficiency [98]. Gamma-retroviral-based vectors require cells to be actively proliferating to gain entry into the cell [99]. Lentiviral-based vectors can infect both cycling and non-cycling cells and carry larger transgenes, but are also limited in their efficiency in primary NK cells as viability is often negatively affected [100]. In order to improve the transduction efficiency of largely virally-resistant NK cells, polybrene, which is a cationic polymer that facilitates viral entry, is commonly used [101]. Another alternative is Retronectin—a truncated version of fibronectin—which works by colocalizing with the virus at the cell’s surface [102,103]. Vectofusin-1, which is a short cationic peptide that functions as a strong transduction enhancer, was originally discovered to greatly enhance the lentiviral transduction of hematopoietic stem cells and has since been adopted in the NK cell field [104]. 

Optimization of the virus packaging identified a higher transduction efficiency for primary NK cells when a Baboon envelope pseudotyped lentivirus (BaEV-LV) was used [105]. The previous standard was vesicular stomatitis virus GP (VSV-G) [106]. BaEV-LV utilizes both ASCT1 and ASCT2 (sodium-dependent neutral amino acid receptors), which are highly expressed on hematopoietic cells, to obtain entry into the cell. The transduction efficiency was also increased when NK cells were previously activated with cytokines prior to transduction, likely as a result of the cytokine-induced proliferation facilitating viral uptake [107,108]. 

### 3.2. Electroporation

One non-viral strategy for genetically engineering NK cells is through the electroporation of DNA or mRNA plasmids. The electroporation of DNA has a low efficiency, but with prior activation of the NK cells combined with mRNA-based plasmids, electroporation efficiencies ranging from 80% to 90% have been achieved in primary NK cells [98,109]. The drawback of using electroporation to obtain CAR expression is the transient expression that results, as no stable integration of the construct in the DNA is possible. The surface expression of CAR constructs on primary NK cells can range from as little as 3-5 days before expression is lost, greatly reducing the therapeutic time window [11,109,110,111]. Electroporation is therefore not a feasible method for genetically engineering a stable CAR-NK cell product. 

### 3.3. Transposons

One alternative for non-viral transduction that is currently being investigated is the use of transposons. DNA transposons are mobile units of DNA, often referred to as ‘jumping genes’, which occur naturally in many organisms. Their ability to randomly integrate throughout the genome makes them an extremely attractive method since viral transduction is often biased towards transcriptionally-active and highly-expressed genes, resulting in their disruption [112]. Two main methods—piggyBac and the sleeping beauty (SB) transposon system—are most commonly used, with an SB-based CAR vector currently being tested in the clinic [75]. SB is originally derived from fish, but retains its activity in the human genome [113]. It consists of two components: the SB transposase mediating the cutting-and-pasting and the DNA vector flanked by terminal inverted repeats (TIRs) to which the transposase can bind to mediate its near random integration into the genome [114]. Genes >100 kb in length can be transduced with this method, making it an attractive higher efficiency option for CAR integration [115].

However, entry of the SB transposase and gene vector into the cell still needs to be facilitated, either through electroporation or viral infection. One method is to use a hybrid adenovirus/transposon vector together with the hyperactive SB100X transposase, which facilitates entry via the adenovirus (non-integrating virus) and DNA integration via the transposase [112]. Another alternative approach is to introduce minicircles containing the gene vector along with transposase mRNA via electroporation [116]. Minicircles are DNA plasmids devoid of the bacterial backbone, which have been shown to have a higher integration efficiency compared to traditional plasmids. The advantage of using transposase mRNA over DNA is the short half-life upon transfer. DNA transposons have the ability to integrate into DNA, but are not stable and have the potential to relocate in the presence of transposases. In order to achieve stable integration, it is advantageous to retain transposase expression for a limited timeframe, allowing for vector integration, but preventing further re-mobilization at a later time point. Furthermore, it eliminates the possibility of the highly-active transposases permanently integrating themselves. 

Apart from the potential re-mobilization of integrated vectors, another limitation to using transposons for CAR transduction is that numerous copies of the CAR construct are introduced per cell. Plasmid-based strategies often result in 8-20 insertions per cell, while a new soluble form of SB transposase produced resulted in only 2-12 insertions per cell [117]. This can lead to excessive activation and potential for NK cell exhaustion, similar to AICD observed in T cells. Nonetheless, transposon-engineered CAR-NK cells are a promising alternative for achieving stable expression without viral integration. 

### 3.4. CRISPR/Cas9

Previously discussed transduction methods have relied on vector integration without specifying the site of integration. The potential for the integration and subsequent disruption of essential survival genes or tumor suppressor genes can be detrimental and dangerous for a clinical product [118]. The discovery of the CRISPR/Cas9 system opened the possibility of targeted gene integration with a high efficiency, compared to similar methods, such as transcription activator-like effector nucleases (TALENs) or zinc-fingers nucleases (ZFNs) [119]. CRISPR/Cas9 utilizes guide RNAs to direct the Cas9 nuclease to the gene of interest to introduce double-stranded breaks. Through homologous directed repair (HDR), a gene vector can be incorporated, providing an efficient method of gene-insertion at a targeted location. CRISPR technology has been used to knock-out the endogenous TCR locus in T cells and replace it with the CAR construct to generate allogeneic universal CAR-T cells [118,120]. In NK cells, targeted CAR insertion combined with gene knock-out opens up new possibilities for improving primary CAR-NK cell products. Targeting key transcription factors involved in NK cell exhaustion, terminal differentiation, and clonal expansion, as observed in adaptive NK cells, and regulators of cell cycle progression and apoptosis, could help in identifying the ideal CAR-NK cell product. 

### 3.5. Cytokine-Based Expansion

Cytokine-based expansion protocols often rely on supra-physiological levels to induce the expansion of primary NK cells. Various combinations of IL-2, IL-12, IL-15, IL-18, and IL-21 have been tested, with the aim of either obtaining maximum cell numbers or tuning the phenotypic profile by expanding a particular subset of NK cells [121]. By combining IL-15 with IL-12 and IL-18, cytokine-induced memory-like NK cells have been generated from mouse and human NK cells, which exhibited an increased anti-tumor response and persistence [122,123,124,125,126,127]. Immature NK cell subsets are most receptive to cytokine input and also possess the highest proliferative capacity, while mature NK cell subsets favor a receptor-mediated input and have a reduced proliferative capacity [90]. High doses of cytokines therefore result in a skewing of the NK cell population towards a more naïve phenotype, but also increase the transcription of effector molecules, including granzyme B and perforin [128]. Expansion protocols solely relying on a cytokine input have a finite fold expansion and often induce cytokine-dependence in the expanded cells. The addiction to supra-physiological levels is a major concern for adoptive cell therapy as the cells undergo massive cytokine-withdrawal upon infusion, severely limiting their in vivo persistence and thus the therapeutic window [129]. The in vitro expansion of primary NK cells with IL-15 has been shown to result in dose-dependent cytokine addiction, which correlated with the degree of proliferation. These cytokine-addicted NK cells were highly susceptible to the induction of apoptosis upon IL-15 withdrawal and this was associated with an altered expression of BCL-2 and BIM—the main anti/pro-apoptotic molecules in cycling NK cells [129]. Cytokine-addiction therefore needs to be considered when infusing an expanded primary NK cell product, with one possible strategy being a gradual reduction in the cytokine concentration prior to infusion. 

### 3.6. Feeder-Based Expansion

Feeder-based expansion protocols combine the receptor-mediated input from the feeder system with cytokine support, to facilitate the large-scale expansion of primary NK cells [121]. A number of different feeder systems have been used, including EBV-transformed lymphoblastoid cell lines (LCLs), autologous peripheral blood mononuclear cells (PBMCs), and various NK cell-sensitive cell lines, such as 721.221 and K562 [130,131,132]. In order to prevent feeder cell outgrowth and facilitate NK cell expansion, feeder cells are irradiated prior to co-culture, effectively preventing further cell division. Elimination of the feeder cells by the NK cells usually occurs within 1-2 weeks, making this a feasible system for clinical application [133]. Compared to cytokine-only-based expansion protocols, feeder-based systems yield higher total fold expansion at lower cytokine doses, making this an attractive expansion method for CAR-based NK cell therapy [121]. Additionally, the lower cytokine doses required for expansion with feeder cells could help mitigate the cytokine withdrawal-induced apoptosis upon infusion.

Genetically engineered feeder cells expressing specific surface ligands to aid in NK cell expansion have been developed. These include membrane-bound cytokines, such as IL-15, IL-18, and IL-21, as well as ligands to activating receptors expressed on NK cells, such as 4-1BBL, HLA-E, and OX40L [132,134,135,136]. While HLA-E-expressing-modified 721.221 cells (721.221-AEH) have been used as a method to preferentially expand adaptive NKG2C^+^ CD56^dim^ NK cells from a bulk NK cell population, 4-1BBL-expressing K562 cells have been used to generally increase NK cell expansion and increase the viability [132,136,137]. Combining K562-4-1BBL with IL-15 and IL-21 support has resulted in superior expansion, reaching up to a 10^5^ fold expansion in 4 weeks, which has been attributed to the IL-21-induced increase in telomere length, prolonging the proliferative capacity of NK cells [137]. Furthermore, the activation of both STAT5 and STAT3 appears to be beneficial in proliferating NK cells. The mechanism by which signaling through 4-1BB augments proliferation still remains to be deciphered. 

### 3.7. In-Vivo Persistence 

Evidence from clinical trials of adoptive cell therapy often fails to detect the presence of infused NK cells more than 1-2 weeks after infusion [91,127,138]. While the short life-span of NK cells contributes to their safety profile in the setting of ACT, it also significantly narrows the therapeutic window, preventing any long-term surveillance by the infused cells. One option for extending the therapeutic window is to give systemic cytokine support to induce proliferation of the infused product in vivo. Although IL-2 and IL-15 have been tested in clinical trials, both are associated with severe side effects [139]. Fever, chills, myalgias, and capillary leak syndrome have been observed after systemic IL-2 treatment, which also has the unwanted side effect of supporting regulatory T cell (T_reg_) expansion [140]. IL-15 does not promote the expansion of any unwanted bystander populations, but has been shown to cause neutropenia when given systemically [141]. Furthermore, systemic IL-15 treatment only induced NK cell expansion for the duration of the treatment, effectively eliminating it as a method for inducing long-term persistence [142]. While optimizing the persistence of CAR-NK cells has definite room for improvement, long-term persistence as observed with CAR-T cells is unlikely. In line with the ACT of activated NK cells, CAR-NK cell therapy has great potential as a bridging therapy for patients with refractory and relapsing malignancies. 

## 4. NK-Specific CAR Constructs

Implementation of the CAR technology in NK cells is often performed with constructs optimized for inducing T cell activation. Although some of the signaling is conserved between T cells and NK cells, namely CD3ζ and 4-1BB, other co-stimulatory domains commonly used in CAR-T cells, as well as TM domains and hinges, are completely absent in NK cells, namely CD8α and CD28 [61,143]. Furthermore, differences in activating signaling, leading to immune synapse formation and subsequent cytolysis of the target cell, are cell type-specific. This is further complicated by the fact that the most potent NK cell subset, the so called ‘serial-killers’, still cannot be identified phenotypically. Here, we will focus on reviewing only NK-specific CAR constructs and the associated signaling in NK cells.

Activating receptors on NK cells utilize a variety of adapter molecules for downstream signaling, including CD3ζ, DAP10, DAP12, and FcRγ chains. While CD3ζ contains and signals via 3 immunoreceptor tyrosine-based activation motif (ITAM) domains, DAP10, DAP12, and FcRγ chains only contain one ITAM domain each. NK cell receptors signaling via CD3ζ include CD16, NKp30, and NKp46, whereby CD16 and NKp46 also signal via FcRγ chains. DAP10 is involved in mediating signaling through NKG2D compared to DAP12, which functions as the adapter protein for activating KIRs, NKG2C, and NKp44 [144]. 

NK-specific CAR constructs have largely focused on incorporating either DAP10 or DAP12 as the activating domain or as a co-stimulatory domain alongside CD3ζ. The incorporation of DAP10 has been successful, but only in combination with NKG2D, in line with endogenous NKG2D signaling. A second-generation CAR construct featuring NGK2D as the ectodomain, with DAP10 and CD3ζ making up the endodomain, exhibited increased surface expression and functionality in primary NK cells tested in vitro and in vivo using an osteosarcoma xenograft mouse model [108]. However, another CAR construct against PD-1 utilizing an NKG2D TM domain fused to 4-1BB and DAP10 found DAP10 to be superfluous, with the same construct lacking DAP10 showing better NK cell functionality [73]. This inhibitory CAR construct aimed to use inhibitory PD-1 signaling to activate the NK cell, utilizing NKG2D only as the TM domain. As a type II protein, NKG2D differs in its transmembrane region when compared to type I membrane proteins that are normally used in CAR constructs, namely CD8α and CD28. The added challenge of using a type II protein for the TM domain was made evident by the poor surface expression of the constructs [73]. Furthermore, ensuring interaction between NKG2D and DAP10 within the constructs appears to be a main factor in determining effective signaling through DAP10 containing-constructs. This was affirmed by another study comparing a CD19-targeting first-generation CAR containing either DAP10 or CD3ζ, where CD3ζ outperformed DAP10 as the activating domain [145]. DAP12, on the other hand, outperformed CD3ζ in first-generation CARs transduced into primary NK cells in both in vitro and in vivo mouse studies, regardless of which scFv was used [109,146].

The largest comparison of NK-specific CAR constructs was performed in NK-92 cells, and included an evaluation of four different transmembrane domains (CD16, NKp44, NKp46, and NKG2D) and four different co-stimulatory domains (2B4, DAP10, DAP12, and CD137) in various combinations with CD3ζ [75]. Only three constructs resulted in an increased cytolytic response, with the construct containing an NKG2D TM domain and 2B4 co-stimulatory domain being chosen for further analysis in iNK cells. Another comparison of 2B4 versus 4-1BB as co-stimulatory domains in NK-92 cells favored the 2B4 containing CAR construct, which induced rapid proliferation, increased cytokine production and degranulation, and decreased apoptosis in the transduced cells [70]. 2B4 is a member of the signaling lymphocytic activation molecule (SLAM)-family receptors and binds CD48, commonly expressed by hematopoietic cells. It signals through its immunoreceptor tyrosine-based switch motif (ITSM), which, upon phosphorylation, recruits adaptor proteins, such as SAP and EAT-2 [147]. 

To address the issue of short in vivo persistence normally associated with infused NK cells, Liu et al. engineered CB-derived NK cells to express IL-15 [148]. The CAR-transduced cells therefore produced their own soluble IL-15, which was sufficient for sustaining autonomous growth over a period of 42 days. However, the infusion of high doses of CAR-expressing CB-derived NK cells into mice (Raji xenograft model) proved fatal in four mice, due to cytokine release syndrome. Pre-empting the possibility of this lethal side effect, the authors also included an inducible suicide gene as a safety measure. While systemic cytokine support upon NK cell infusion is unfavorable due to the side effects of both IL-2 and IL-15, engineering cells to secrete cytokines does not appear to be a simple solution either. 

### 4.1. Clinical Trials of CAR-NK Cells

Clinicaltrials.gov currently has six CAR-NK cell trials listed as actively recruiting and six early Phase I clinical trials that are not yet actively recruiting (Table 1). While many trials are focusing on lymphoma and leukemia, others are targeting solid tumors, including ovarian, prostate, brain, liver, lung, and pancreatic cancer. 

Results from small-scale clinical trials (*n* = 3) with both NK-92 and primary CAR-NK cells targeting CD33 or NKG2D ligands have been reported [109,149], but the first large-scale Phase I/II clinical trial was only recently published in February 2020 [89]. Eleven patients with either relapsed or refractory chronic lymphocytic leukemia (CLL) or non-Hodgkin’s lymphoma received an allogeneic CB-derived CAR-NK cell product after undergoing a standard lymphodepleting treatment of cyclophosphamide/fludarabin. Although donor NK cells were originally chosen based on a partial HLA-match (4/6), the absence of GvHD resulted in donor criteria focusing on KIR-ligand mismatch instead, with no regard given to HLA-matching for the final two patients. Unfortunately, the number of donors receiving a KIR-ligand mismatched product was too low (5/11) to draw any conclusions. Eliminating the need for HLA-matching highlights the possibility of generating a truly off-the-shelf product, although the viability and potency of the product after a freeze/thaw cycle still need to be clinically tested. The short manufacturing time of the CAR product enabled each patient to receive an individually manufactured clinical product within 2 weeks of enrollment into the clinical study. Eight out of the 11 patients responded to the treatment, with seven patients achieving complete remission. The high response rate and absence of serious side effects, such as CRS, GvHD, and neurotoxicity, proved the feasibility and efficacy of CAR-NK cells as promising new cancer immunotherapy. 

Compared to the previously published in vitro study, where increased levels of IL-15 were detected in the supernatant of the IL-15-producing CAR-NK cells sustaining autonomous cell growth, serum levels of IL-15 in treated patients did not exceed baseline levels [89,148]. The detection of CAR-NK cells in circulation by flow cytometry was limited to the first 14 days and highly variable among donors. Quantitative PCR was used for long-term detection of the vector transgene, although this only correlated with the treatment dose received for the first 14 days. While the durability of the CAR-NK cell therapy could not be assessed, as remission consolidation therapy was allowed after the initial 30 days, patients that responded to the therapy exhibited a significantly higher early expansion of CAR-NK cells. Considering the severity of disease and multiple rounds of failed chemotherapy (3-11) these patients had previously undergone, a response rate of 8 out of 11 patients is a tremendous success.

### 4.2. Endogenous Signaling in CAR-NK Cells

Inhibitory receptor ligation by self MHC-I molecules fine-tunes the functional potential of an NK cell through modulation of the lysosomal compartment, leading to granzyme B retention in cytotoxic granules [150]. Educated NK cells, having received an inhibitory receptor input from cognate ligands, exhibit an increased functional potential upon receiving an adequate activating receptor input compared to uneducated NK cells. The main inhibitory receptors educating naïve NK cells are NKG2A and KIRs. NKG2A-mediated inhibition is eventually replaced by the stronger KIR-mediated inhibition during maturation [151]. Oei et al. have addressed the question of whether or not CAR signaling was strong enough to overcome the endogenous inhibitory signaling [11]. Indeed, CAR-expressing NKG2A^+^ NK cells were able to overcome HLA-E mediated inhibition and effectively lyse 721.221-AEH cells. However, this was not the case for KIR-mediated inhibition, whereby cognate self-ligand expression on tumor cells dampened the cytolytic response of CAR-expressing NK cells. While CAR expression increased the functional response to antigen-expressing targets cells, the functional hierarchy between educated and uneducated cells was maintained [11]. Hence, the selection of a functional NK cell starting population is highly advantageous for maximizing the anti-tumor effect. 

## 5. Perspective on the Future of CAR-NK Cells

The success of CAR-T cell therapy against CD19-expressing lymphomas in the clinic has facilitated rapid progression in the CAR-NK cell field. FDA approval of the first genetically modified cell product has paved the way to the clinic for CAR-NK cells, but simply incorporating constructs optimized for T cells into NK cells is suboptimal. The biological and molecular mechanisms leading to cellular activation greatly differ between T and NK cells and thus need to be considered when designing a CAR-NK cell construct. Combination therapy of CD16-expressing CAR-NK cells together with monoclonal antibody therapy is one possibility for utilizing the full cytotoxic potential of NK cells through both target-specific lysis and ADCC. The challenges of genetically engineering primary NK cells has resulted in many studies and clinical trials being performed with NK-92 cells. The development of new methods and optimization of existing technologies will facilitate more efficient genetic engineering of primary NK cells, allowing primary NK cells to replace the largely NK-92 CAR products currently being tested in the clinic. 

Lastly, if the problem of persistence can be addressed, either through additional engineering of the construct or by furthering our understanding of the requirements for sustained NK cell proliferation, CAR-NK therapy will have great potential as a new immunotherapy against cancer. 

## Figures and Tables

**Figure 1 cancers-12-00706-f001:**
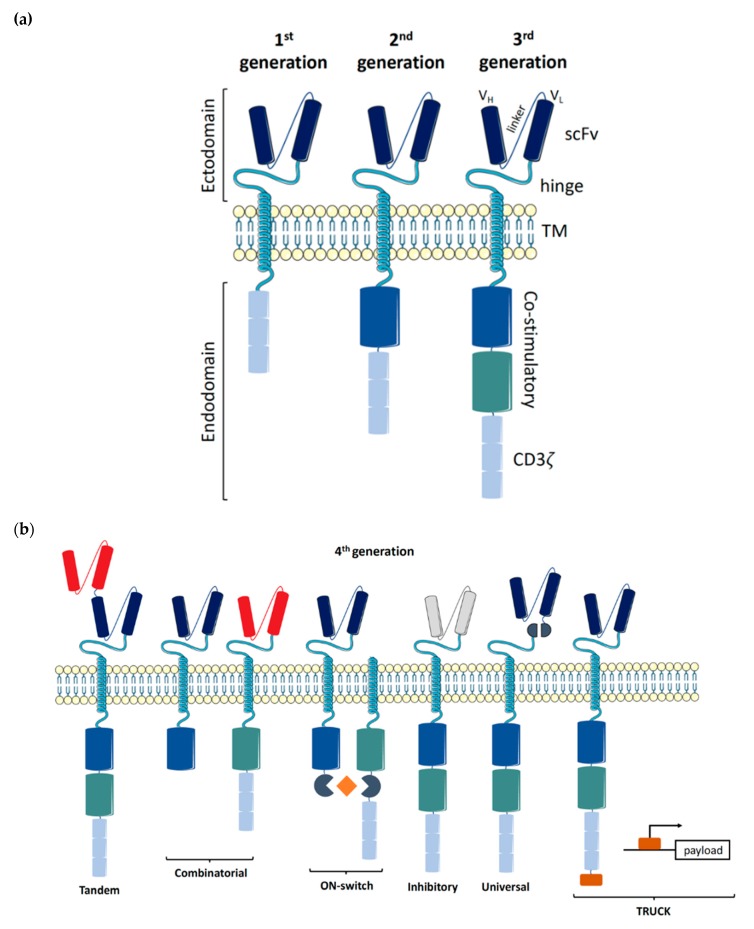
Evolution of chimeric antigen receptor (CAR)-T cell constructs. The main components of a chimeric antigen receptor and its evolution over time. (**a**) 1st, 2nd and 3rd generation CAR constructs; (**b**) Examples of 4th generation CAR construct organized by subgroups. Abbreviations: scFv, single-chain variable fragment; TM, transmembrane; TRUCK, T cells redirected for universal cytokine killing.

**Figure 2 cancers-12-00706-f002:**
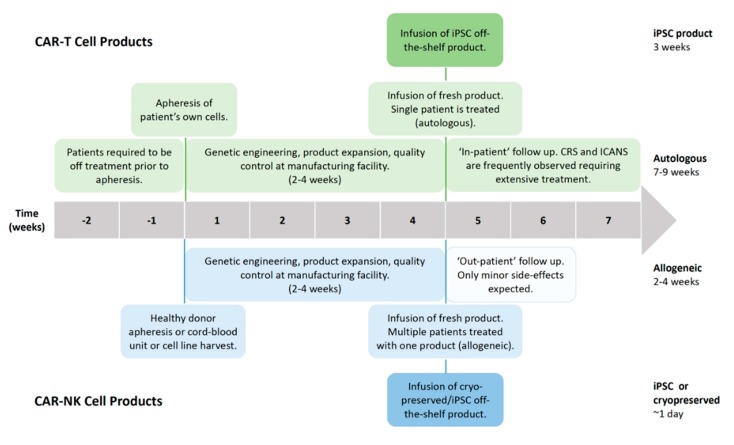
Manufacturing timeline of CAR-T cell and CAR-natural killer (NK) cell products. Comparison of the manufacturing time for CAR-T and CAR-NK cell products, from harnessing the cells for product processing to patient monitoring after treatment. Abbreviations: CRS, cytokine release syndrome; ICANS, immune effector cell-associated neurologic syndrome; iPSC, induced pluripotent stem cell.

**Figure 3 cancers-12-00706-f003:**
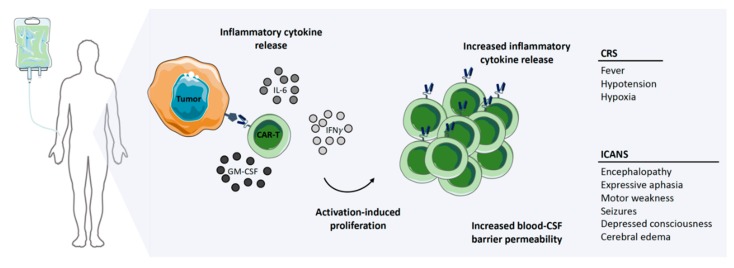
CAR-T cell treatment-related toxicities. Toxicities associated with the antigen-induced activation and proliferation of CAR-T cells caused by the release of cytokines. Abbreviations: IL-6, interleukin-6, IFNγ, interferon gamma; GM-CSF, granulocyte-macrophage colony-stimulating factor; CSF, cerebrospinal fluid; ICANS, immune effector cell-associated neurologic syndrome; CRS, cytokine release syndrome.

**Figure 4 cancers-12-00706-f004:**
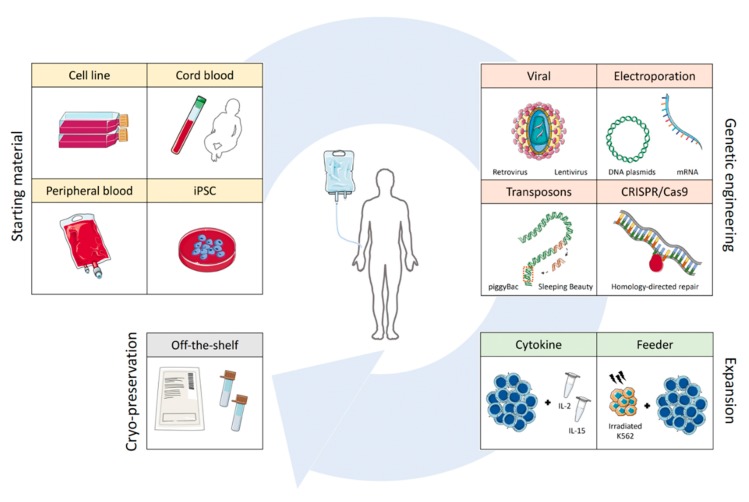
Overview of the CAR-NK cell manufacturing process. The four manufacturing stages of CAR-NK cell production, from the choice of starting material to genetic modification, product expansion, and the subsequent cryopreservation of treatment doses. Abbreviations: iPSC, induced pluripotent stem cell; CRISPR, clustered regulatory interspaced short palindromic repeats; IL-2, interleukin-2; IL-15, interleukin-15.

**Table 1 cancers-12-00706-t001:** Current clinical trials of CAR-NK cells. Clinical trials featuring CAR-NK cells against various target antigens and diseases which are currently actively recruiting or are scheduled to begin recruiting in the near future. Data was obtained from clinicaltrials.gov. *Abbreviations: iPSC, induced pluripotent stem cell*.

Identifier	Status	Disease	Target	Source	Location/Sponsor	Phase
**NCT03415100**	Recruiting	Metastatic Solid Tumours	NKG2D ligands	PB NK cells	Third Affiliated Hospital of Guangzhou Medical University, Guangzhou, Guangdong, China	Phase I
**NCT03940820**	Recruiting	ROBO1+ Solid Tumors	ROBO1	NK-92	Suzhou Hospital Affiliated to Nanjing Medical University, Suzhou, Jiangsu, China	Phase I/II
**NCT03940833**	Recruiting	Multiple Myeloma	BCMA	NK-92	Wuxi People’s Hospital, Wuxi, Jiangsu, China	Phase I/II
**NCT03941457**	Recruiting	ROBO1+ Pancreatic Cancer	ROBO1	NK-92	Shanghai Ruijin Hospital, Shanghai, China	Phase I/II
**NCT03056339**	Recruiting	B-Lymphoid Malignancies, Acute Lymphocytic Leukemia, Chronic Lymphocytic Leukemia, Non-Hodgkin Lymphoma	CD19	CB NK cells	University of Texas MD Anderson Cancer Center, Houston, Texas, United States	Phase I/II
**NCT03383978**	Recruiting	HER2+ Glioblastoma	HER2	NK-92	Johann W. Goethe University Hospital, Frankfurt, Germany	Phase I
**NCT03692767**	Not yet recruiting	Refractory B-Cell Lymphoma	CD22	Unknown	Allife Medical Science and Technology, Beijing, China	Early Phase I
**NCT03690310**	Not yet recruiting	Refractory B-Cell Lymphoma	CD19	Unknown	Allife Medical Science and Technology, Beijing, China	Early Phase I
**NCT03692637**	Not yet recruiting	Epithelial Ovarian Cancer	Mesothelin	PB NK cells	Allife Medical Science and Technology, Beijing, China	Early Phase I
**NCT03692663**	Not yet recruiting	Castration-resistant Prostate Cancer	PSMA	Unknown	Allife Medical Science and Technology, Beijing, China	Early Phase I
**NCT03824964**	Not yet recruiting	Refractory B-Cell Lymphoma	CD19/CD22	Unknown	Beijing Cancer Hospital, Beijing, China	Early Phase I
**NCT04245722**	Not yet recruiting	B-Cell Lymphoma, Chronic Lymphocytic Leukemia	CD19	iPSC	Fate Therapeutics, San Diego, USA	Phase I

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
