# Peer review of "You Have Got a Fast CAR: Chimeric Antigen Receptor NK Cells in Cancer Therapy"

_cancers, 2020, doi:10.3390/cancers12030706_

Round 1
Reviewer 1 Report
The manuscript by Pfefferle and Huntington is a review article about the current and potential use of CAR-NK cells as an alternative to CAR-T cell immunotherapies. The manuscript is mostly well-balanced and provides an informative overview of the CAR-NK field. Below are a few minor points that the authors should address before this manuscript is ready for publication.
The authors should also introduce iPSC-derived T cells (CAR-T cells) to provide a balanced overview of this possibility in comparison to iPSC-NK cells (CAR-NK cells).
The NK cell modification section could be improved by including methods such as use of vectofusin or other molecules to increase viral vector transduction efficiency.
The authors use a mix of British and American English style througout the manuscript (e.g. tumour, leukemia, haematopoietic, etc.), this should be reconciled to one style.
The authors should carefully proofread the manuscript to eliminate minor language mistakes, including “remain to be address” should be addressed, “10% of currently clinical trials” should be “current”, scFv sometimes written scFV as well as many others.
Author Response
The manuscript by Pfefferle and Huntington is a review article about the current and potential use of CAR-NK cells as an alternative to CAR-T cell immunotherapies. The manuscript is mostly well-balanced and provides an informative overview of the CAR-NK field. Below are a few minor points that the authors should address before this manuscript is ready for publication.
Response: We thank the reviewer for their positive comments.
The authors should also introduce iPSC-derived T cells (CAR-T cells) to provide a balanced overview of this possibility in comparison to iPSC-NK cells (CAR-NK cells). …..
Response: We have now introduced iPSC-derived CAR T cells in section 1.5 of the text as well as in Figure 2.
The NK cell modification section could be improved by including methods such as use of vectofusin or other molecules to increase viral vector transduction efficiency.
Response: We have now expanded section 3.1 of the text where we discuss the use of polybrene, Retronectin and Vectofusin-1 in viral transduction of NK cells. Furthermore, we have expanded the virus packaging methods section to include VSV-G as well as BaEV-LV.
The authors use a mix of British and American English style througout the manuscript (e.g. tumour, leukemia, haematopoietic, etc.), this should be reconciled to one style.
Response: We have previously used Australian English through the manuscript, but due to the confusion noted by the reviewer, we have now switched to American English.
The authors should carefully proofread the manuscript to eliminate minor language mistakes, including “remain to be address” should be addressed, “10% of currently clinical trials” should be “current”, scFv sometimes written scFV as well as many others.
Response: We thank the reviewer for pointing out these language mistakes. We have amended these mistakes as well as others identified.
Reviewer 2 Report
Aline et al reviewed the CAR-NK field, while I can't get too much new ideas from this manuscript. I don't think it is well enough to be published in Cancers at present.
As a review, the contents were not substantial, and lack of novel points that can catch readers' view.
Acturally, evolution of CAR has come to 4th, even 5th generation, Fig 1 is really old and too simple.
The quality of the figures in whole manuscript should be strengthened a lot, also are short of novel perspective.
Author Response
Aline et al reviewed the CAR-NK field, while I can't get too much new ideas from this manuscript. I don't think it is well enough to be published in Cancers at present.
As a review, the contents were not substantial, and lack of novel points that can catch readers' view.
Acturally, evolution of CAR has come to 4th, even 5th generation, Fig 1 is really old and too simple.
The quality of the figures in whole manuscript should be strengthened a lot, also are short of novel perspective.
Response: We have added substantial content to the majority of the sections in the manuscript and improved all four figures, as well as added a new table with clinical trial information. Figure 1 now includes an overview of the next generation or 4th generation CARs which are still largely limited to the CAR-T cell field. While we agree that there have been a number of recent reviews focusing on the CAR field, and in particular on the CAR-NK cell field, we believe that by focusing on the NK cell specific signalling aspect and the manufacturing process, in particular differences between cytokine and feeder based expansion we have added novelty to this manuscript. We hope that the extra 20% of text we have added to the manuscript add novelty to this review of CAR-NK cells.
Reviewer 3 Report
Very well written and comprehensive manuscript
The prospects from CARNK cells are very well presented. Below there are few suggestions for further improving the quality of the manuscript.
1) Section 1.4 The paragraph ''Successful clinical trials with 1st generation autologous CD19-targeted CAR-T cells resulted in FDA approval, with two products, namely Kymriah® (Novartis) and Yescarta® (Kite/Gilead)
currently on the market'' is not correct. These are 2nd generation constructs
2) Section 1.4 Paragraph ''a high fever to life-threatening capillary leakage tachycardia requiring intensive care treatment'' should be corrected
3) Again in Section 1.4 Paragraph ''including IL-6, IFN, GM-CSF and soluble IL-6 an IL-2 receptor'' should be corrected
4) Section 1.5 Paragraph ''CAR-NK cells, on the other hand, retain their innate cytolytic capacity against germline-encoded tumour/stress ligands
and are able to detect MHC-I negative tumour cells due to their lack of self-antigen'' should be eliminated because it is not correct.
Although in general it is correct that NK can destroy cells without MHC expresion this is not the case in CAR-NK cells where cytotoxicity is mediated through the chimeric antigen receptor. So loss of antigen will make also CAR-NK ineffective
5) Section 2.1 Authors need to provide proper references that irradiation does not block the cytolytic activity of NK cells
Irradiated mononuclear cells express significant in vitro cytotoxic activity: promise for in vivo clinical efficacy of irradiated mismatched donor lymphocytes infusion. Tsirigotis P, Resnick IB, Kapsimalli V, Dray L, Psarra E, Samuel S, Spyridonidis A, Konsta E, Vikentiou M, Or R, Slavin S, Shapira MY. Immunotherapy. 2014;6(4):409-17
6) Authors should provide data from animal data or clinical trials regarding the efficacy and safety of CAR-NK,
Author Response
Very well written and comprehensive manuscript
The prospects from CARNK cells are very well presented. Below there are few suggestions for further improving the quality of the manuscript.
Response: We thank the reviewer for their positive comments.
1) Section 1.4 The paragraph ''Successful clinical trials with 1st generation autologous CD19-targeted CAR-T cells resulted in FDA approval, with two products, namely Kymriah® (Novartis) and Yescarta® (Kite/Gilead) currently on the market'' is not correct. These are 2nd generation constructs
Response: We thank the reviewer for noting this mistake. It has now been corrected.
2) Section 1.4 Paragraph ''a high fever to life-threatening capillary leakage tachycardia requiring intensive care treatment'' should be corrected
Response: We thank the reviewers for this comment and have now corrected the corresponding section in the manuscript.
3) Again in Section 1.4 Paragraph ''including IL-6, IFNg, GM-CSF and soluble IL-6 an IL-2 receptor'' should be corrected
Response: We have amended this statement and added more references to support it.
4) Section 1.5 Paragraph ''CAR-NK cells, on the other hand, retain their innate cytolytic capacity against germline-encoded tumour/stress ligands and are able to detect MHC-I negative tumour cells due to their lack of self-antigen'' should be eliminated because it is not correct.
Although in general it is correct that NK can destroy cells without MHC expresion this is not the case in CAR-NK cells where cytotoxicity is mediated through the chimeric antigen receptor. So loss of antigen will make also CAR-NK ineffective
Response: We disagree with the reviewer’s statement. We agree that the introduction of a CAR construct increases the cytotoxic capacity of the NK cell and redirects it to the CAR-specific antigen. However, the CAR-NK cells endogenous cytotoxic capacity is not restricted to the CAR construct or by the introduction of a CAR construct. Although cord-blood derived NK cells have limited cytotoxicity at resting state, cytokine-based expansion during the manufacturing process increases their cytotoxicity. Oei et al have shown that mock transfected and CD19-CAR NK cells kill K562 (missing MHC-I ligands) equally well and it is only in the presence of CD19+ cell lines that increased cytotoxicity of CAR-NK cells is shown. Furthermore, Liu et al have in their recent clinical trial aimed to select KIR-ligand mismatch cord-blood donors for the same reason, to increase endogenous NK cell killing in the absence of self-ligand.
5) Section 2.1 Authors need to provide proper references that irradiation does not block the cytolytic activity of NK cells
Irradiated mononuclear cells express significant in vitro cytotoxic activity: promise for in vivo clinical efficacy of irradiated mismatched donor lymphocytes infusion. Tsirigotis P, Resnick IB, Kapsimalli V, Dray L, Psarra E, Samuel S, Spyridonidis A, Konsta E, Vikentiou M, Or R, Slavin S, Shapira MY. Immunotherapy. 2014;6(4):409-17
Response: We thank the reviewer for the reference and have now included it in the manuscript along with others studies showing efficacy of NK-92 CAR-NK cells in vitro and in vivo.
6) Authors should provide data from animal data or clinical trials regarding the efficacy and safety of CAR-NK,
Response: We have now added section 4.1 where we discuss the recently published clinical trial of CB-derived CAR-NK cells in 11 patients. Furthermore, we have added Table 1 which outlines currently ongoing clinical trials as well as future trials already listed on clinicaltrials.gov. We have also mentioned in vivo mouse data for a number of specific constructs discussed throughout the manuscript.
Round 2
Reviewer 2 Report
Authors added lots of contents in revised manuscript, while the quality of the figures are still too low, looks like copies, also table 1 is so dim.
Author Response
We respectively decline to modify the figures further but have modified the table by using brighter shades of blue in line with the reviewer’s comments.